# Comparative Metabolome Analyses of Ivermectin-Resistant and -Susceptible Strains of *Haemonchus contortus*

**DOI:** 10.3390/ani13030456

**Published:** 2023-01-28

**Authors:** Waresi Tuersong, Xin Liu, Yifan Wang, Simin Wu, Peixi Qin, Shengnang Zhu, Feng Liu, Chunqun Wang, Min Hu

**Affiliations:** State Key Laboratory of Agricultural Microbiology, Key Laboratory of Development of Veterinary Products, Ministry of Agriculture, College of Veterinary Medicine, Huazhong Agricultural University, Wuhan 430070, China

**Keywords:** *Haemonchus contortus*, ivermectin resistance, metabolomics

## Abstract

**Simple Summary:**

Ivermectin (IVM) is a highly effective, broad-spectrum, gold-standard antiparasitic drug that has been used extensively to control nematodes such as *Haemonchus contortus*. However, IVM-resistant *H. contortus* populations are now common throughout the world, and there is limited understanding of the mechanisms underlying IVM resistance in *H. contortus*. Recent studies have offered evidence that the metabolic status of bacteria significantly affects their antibiotic sensitivity and that specific metabolic characteristics are related to resistance. Thus, in the present study, we explored the metabolome of an IVM-resistant *H. contortus* strain and analyzed the role of metabolism in IVM resistance. These results contribute to our understanding of the mechanisms of IVM resistance in *H. contortus*.

**Abstract:**

Resistance to anthelmintics such as ivermectin (IVM) is currently a major problem in the treatment of *Haemonchus contortus*, an important parasitic nematode of small ruminants. Although many advances have been made in understanding the IVM resistance mechanism, its exact mechanism remains unclear for *H. contortus*. Therefore, understanding the resistance mechanism becomes increasingly important for controlling haemonchosis. Recent research showed that the metabolic state of bacteria influences their susceptibility to antibiotics. However, little information is available on the roles of metabolites and metabolic pathways in IVM resistance of *H. contortus*. In this study, comparative analyses of the metabolomics of IVM-susceptible and -resistant adult *H. contortus* worms were carried out to explore the role of *H. contortus* metabolism in IVM resistance. In total, 705 metabolites belonging to 42 categories were detected, and 86 differential metabolites (17 upregulated and 69 downregulated) were identified in the IVM-resistant strain compared to the susceptible one. A KEGG pathway analysis showed that these 86 differential metabolites were enriched in 42 pathways that mainly included purine metabolism; the biosynthesis of amino acids; glycine, serine, and threonine metabolism; and cysteine and methionine metabolism. These results showed that amino acid metabolism may be mediated by the uptake of IVM and related with IVM resistance in *H. contortus.* This study contributes to our understanding of the mechanisms of IVM resistance and may provide effective approaches to manage infection by resistant strains of *H. contortus*.

## 1. Introduction

Infection by *Haemonchus contortus*, a highly pathogenic and economically important parasitic nematode, is ubiquitous in small grazing ruminants in warm climatic regions [1,2]. *H. contortus* infection is a major health problem in small ruminants, causing edema, anemia, and sudden death in infected sheep and goats, especially in young animals with heavy worm burdens [3]. Due to the high rates of infection and mortality in small ruminants around the world, small ruminant farmers have sustained significant economic losses [4]. The haemonchosis caused by *H. contortus* has been treated or prevented over the last 50 years using broad- or narrow-spectrum anthelmintics, such as ivermectin (IVM) and albendazole (ALB), with tremendous benefits to livestock health and breeding [5,6]. Among them, ivermectin (IVM) is one of the highly effective drugs used for the treatment and prevention of infections of parasitic endo- and ectoparasites in humans and animals. IVM is one of the most successful forms of anthelmintics treatment in the history of medicine and has improved the lives of many millions of peoples and animals. The researchers who discovered ivermectin won the Nobel Prize in 2015 [7]. Regrettably, the successful use of any anthelmintics is compromised by the rapid appearance of resistance. Similar to ALB, the inappropriate use and/or misuse of IVM has promoted the emergence and rapid spread of resistance in *H. contortus*, and reports of its resistance are no longer noteworthy, posing a serious threat to livestock and human health [8,9,10,11,12]. For this reason, the mechanisms of IVM resistance have been extensively studied in *H. contortus*.

Although numerous investigations of the IVM resistance mechanisms in *H. contortus* have been carried out, the major genetic mediators that can explain IVM resistance in most field isolates of *H. contortus* have not been unequivocally identified [13]. Previous research demonstrated several mechanisms by which *H. contortus* develops IVM resistance, including mutations or changes in the expression levels of IVM target genes, the induction or upregulation of drug efflux, and changes in drug uptake pathways [14,15,16,17]. For example, mutations in GluCl genes (avr-14, avr-15, and glc-1) confer high-level resistance to the antiparasitic drug ivermectin in *H. contortus* and *C. elegans* [18]. However, the complexities of the inherent characteristics of *H. contortus*, such as the extremely high levels of genetic diversity within populations, still impede efforts to overcome IVM resistance in *H. contortus*, requiring innovative approaches [19]. Thus, it is necessary to understand the IVM resistance mechanisms in *H. contortus*.

Although the mechanisms of action and primary targets for antibiotics and anticancer drugs are well studied, there is growing evidence and appreciation that altered metabolism affects drug efficacy, and this has been recognized as a key determinant of drug resistance in cancer [20]. For example, amino acid metabolism supports the cellular demand to overcome drug-induced stress [20]. Furthermore, recent intense studies have shown that the therapeutic pressure of antibiotics leads to metabolic alterations and drives resistance to antibiotics in bacteria [21]. For example, bacteria with reduced amino acid metabolism tend to be tolerant or resistant to conventional antibiotics due to lower intracellular antibiotic accumulation [22]. Therefore, there is a meaningful lesson that may apply to the study of IVM resistance mechanisms in *H. contortus*. IVM may also alter the metabolic state of *H. contortus,* and the metabolic states of resistant strains influence their susceptibility to IVM. Here, we analyzed the metabolic states of IVM-resistant and -susceptible strains using LC-MS-based metabolomics. These results increase our understanding of IVM resistance and may provide a new theoretical basis for IVM resistance mechanisms.

## 2. Materials and Methods

### 2.1. H. contortus Strains and Samples

Two isolates of *H. contortus* were used in this study:(i)The Haecon-5 strain kindly presented by Professor Robin B. Gasser (University of Melbourne) and maintained in goats in Huazhong Agricultural, which is sensitive to all commercial anthelmintics.(ii)The Zhaosu strain, which was originally isolated in Zhaosu County, Xinjiang, China, as an IVM-resistant strain [22].(iii)Samples: Six goats (6-month-old goats free of parasites) were randomly divided into two groups (resistant strain and susceptible strain, three goats/group) and infected with the infective third-stage larvae (7000 L3s per goat) of the respective strain. The adult male and female worms of *H. contortus* were isolated from the abomasa of goats 45 days after infection. These adult worms were thoroughly washed in PBS and assigned into two groups with three biological replicates for each group. Each sample was made from 160 adult worms (80 males and 80 females/sample) of Haecon-5-S or Zhaosu-R. All samples were transferred to liquid nitrogen for storage until use.

### 2.2. Metabolite Extraction and HPLC-MS/MS Analysis

Worms (100 mg/sample) collected from different goats were prepared and extracted following the procedures of a previous study [23]. After extraction, the samples were loaded into an ExionLC™ AD system (SCIEX) coupled with a QTRAP ^®^ 6500+ mass spectrometer (SCIEX) at Novogene Co., Ltd. (Beijing, China). As metabolites are susceptible to interference from external factors and change rapidly, quality control during the assay was particularly important for the accuracy of the results. In this experiment, the mass spectrometry process was divided into quality control (QC) samples and experimental samples for testing. The QC (quality control) samples were made of equal volumes of experimental samples that were mixed to evaluate the stability of the chromatography–mass spectrometry system throughout the experimental process and to perform data quality control. The raw data generated by LC-MS/MS were analyzed, and metabolites were quantified and identified using the Q1, Q3, RT (retention time), DP (declustering potential), and CE (collision energy). All parameters and conditions for LC−MS/MS analyses were similar to the previous study [24].

### 2.3. Data Analysis

A principal components analysis (PCA) and a partial least-squares discriminant analysis (PLS-DA) were performed using metaX. In order to estimate the quality of the PLS-DA model, the model was sorted and verified to check whether the model was “over fitted”. The estimate standard was that when the R2 value was greater than the Q2 value and the intercept between the Q2 regression line and the Y axis was less than 0, the model was not “over fitted”. Prior to the differential analysis of the total identified metabolites, the data were log-transformed and normalized using the metaX software to obtain VIP values for each metabolite. We applied three parameters to identify the statistical differential metabolites. The VIP (variable importance in the projection) values indicate the contributions of metabolites to the grouping. FC refers to the fold change, which is the ratio of the mean values of all biological replicate quantification values for each metabolite in the comparison group. These differential metabolites were identified with VIP > 1, *p*-value < 0.05, and FC ≥ 1.5 or FC ≤ 0.667, and R software was used to draw the volcano plots and clustering heat map. The potential roles of these metabolites were studied using the KEGG database (http://www.genome.jp/kegg/ (accessed on 26 November 2022)), and the metabolic pathways were considered to have statistically significant enrichment when the *p*-values of the metabolic pathways were <0.05.

## 3. Results

### 3.1. Data Analysis for Quality Control of Samples

The results showed that the total ion flow pattern of QC samples was of high quality with three replicates in the positive and negative ion modes. The intensity and retention times of the spectral peaks showed high reproducibility, indicating good instrument stability during the detection process (Figure 1A,B). In addition, the correlation analysis of the QC samples was performed based on the relative quantitative values of metabolites. From Figure 1C, the correlation coefficients of the QC samples were above 0.994, indicating better stability of the whole assay process and higher data quality. The above results demonstrated that the method used in this test was stable and reproducible and that the raw data that were collected met the requirements of the subsequent analysis.

### 3.2. Total Sample PCA and PLS-DA Analyses

The principal component analysis (PCA) results showed high clustering of QC samples and good separation between the susceptible and resistant strains, indicating the good stability and high data quality of the whole method (Figure 2A). In addition, the quality of the model was evaluated with a PLS-DA analysis, and the results showed that the R2 and Q2 values were found to be 1 and 0.51, respectively. The intercept of the Q2 regression line with the *Y*-axis was less than 0, indicating that the OPLS-DA model was not over-fitted and was valid (Figure 2B). The above results indicated that the accuracy and reliability of the obtained data were high enough for the subsequent analysis.

### 3.3. Overall Metabolomics Analysis of H. contortus

As shown in Table 1 a total of 705 metabolites belonging to 42 categories were identified in the resistant and susceptible strains, including 154 amino acids and derivatives, 105 organic acids and derivatives, 74 nucleotides and derivatives, 50 fatty acyls, 42 carbohydrates and derivatives, 36 hormones, 26 phospholipids, 22 carnitines, 20 organic heterocyclic compounds, 14 sugar acids and derivatives, 13 vitamins, 12 oxidized lipids, 11 others, 11 bile acids, 11 benzoic acid and its derivatives, 10 indole and its derivatives, 10 benzene and its derivatives, and other metabolites with less than 10 metabolites (Table 1 and Appendix A).

### 3.4. Differential Metabolite Analysis

In this study, differential metabolites between IVM-resistant and -susceptible *H. contortus* strains were identified by combining the variables with the VIP, FC, and *p*-values. In addition, a cluster analysis and an ROC curve analysis were performed to evaluate differential metabolites. The results showed that 86 differential metabolites were identified in the IVM-resistant strain compared to the susceptible strain. Among them, 17 metabolites were upregulated and 69 were downregulated in the IVM-resistant strain (Figure 3A). The types of differential metabolites mainly included amino acids and derivatives (28), organic acids and derivatives (14), nucleotides and derivatives (10), hormones (5), carnitines (5), organic heterocyclic compounds (3), fatty acyl groups (3), vitamins (3), phenols and derivatives (2), sugar alcohols (2), etc. (Appendix A). The results indicate that the metabolic profiles of the resistant strain were substantially different from those of the susceptible strain. The corresponding ROC curve of the differential metabolites had an AUC of 1, indicating that the identification accuracy of the differential metabolites was very high (Figure 3B). To assess the relationships among the differential metabolites, all differential metabolites were used for a hierarchical cluster analysis (Figure 4), showing that the differential metabolites from the biological replicates of the same strain were more closely clustered together. These results further confirmed the high accuracy and reproducibility of the experimental results.

### 3.5. Functional Annotation Analysis of the Differential Metabolites

The altered metabolites in the resistant strain were analyzed, and the significantly enriched pathways are displayed in the scatter diagram (Figure 5 & Appendix A). The results showed that 86 differential metabolites were enriched in 43 pathways that mainly included purine metabolism; the biosynthesis of amino acids, specifically glycine, serine, and threonine metabolism; cysteine and methionine metabolism; and tryptophan metabolism. In addition, riboflavin metabolism was also significantly changed.

## 4. Discussion

Although the main targets and mechanisms of action for antibiotics are well studied, there is growing evidence that altered metabolism actively participates in antibiotic efficacy [25] and that extracellular metabolites may either potentiate [26,27] or suppress [28] the killing activities of the antibiotics. Here, based on the study of bacterial antibiotic resistance, we analyzed the metabolomes of IVM-resistant and IVM-sensitive *H. contortus* using LC-MS and identified some metabolites and pathways that could play important roles in IVM resistance in *H. contortus*. Importantly, these results could provide a new idea for the study on the mechanism of IVM resistance in *H. contortus* and related parasitic nematodes.

Metabolomics has been used to study the metabolic characteristics in antibiotic-resistant bacteria. For example, in kanamycin-resistant *Edwardsiella tarda*, decreased abundances of alanine, glutamate, glucose, and fructose were detected, and exogenous additions of these metabolites promoted antibiotic uptake and restored the susceptibility of resistant *E. tarda* bacteria [29]. In addition, in apramycin-resistant *Salmonella*, the metabolites citrulline and glutamine were significantly reduced, and the killing effect of apramycin was restored when the two metabolites were added [30]. Furthermore, the synergistic use of glycine, serine, and threonine elevated the intracellular kanamycin concentration and increased the lethal efficacy of kanamycin [31]. The underlying mechanism of the above studies is that exogenous amino acids promote TCA cycle substrate activation and increase the concentration of NADH. The NADH is oxidized through the electron transport chain to generate proton motive force (PMF), and PMF facilitates the antibiotic uptake [29]. In the present study, we noted that amino acids and derivatives were significantly decreased in the IVM-resistant strain, namely 24 amino acids and derivatives were downregulated and 4 were upregulated. Interestingly, among them, alanine, citrulline, glutamine, and glycine were significantly downregulated in the IVM-resistant strain. Moreover, the differential amino acids and derivatives were significantly enriched in the biosynthesis of amino acids; glycine, serine, and threonine metabolism; cysteine and methionine metabolism; and the tryptophan metabolism pathway, suggesting that amino acids and derivatives may play significant roles in IVM resistance, although this remains to be verified.

A recent study showed that the production of reactive oxygen species (ROS) was recognized as a common mechanism for antibiotic killing [32]. For example, Ye et al. showed that exogenous alanine could promote ROS production that in turn potentiated the killing of antibiotic-resistant *E. tarda*. The underlying mechanism was that alanine metabolism could connect to riboflavin metabolism, which was the the source for ROS production [33]. Intriguingly, alanine was one of the significantly downregulated metabolites, and a functional annotation analysis of the differential metabolites showed that riboflavin metabolism was significantly enriched. Therefore, we speculate that the alanine in *H. contortus* influences the killing efficacy of IVM via connecting with riboflavin metabolism. Furthermore, growing evidence shows that amino acid metabolism has a pleiotropic role in drug resistance in cancer cells and drives drug resistance by sustaining redox homeostasis, regulating epigenetic modification, maintaining biosynthetic processes, and offering metabolic intermediates for energy generation [20,34,35]. For instance, enhanced glutamine metabolism supports the survival of resistant cancer cells via the NADPH-dependent glutathione redox system, and the underlying mechanism is that glutamine-derived antioxidants protect against the oxidative stress prompted by drug-induced ROS [16]. Therefore, the above results indicated that amino acid metabolism should be considered a critical determinant of the sensitivity of *H. contortus* to IVM.

In addition, a recent study showed that perturbations in purine biosynthesis alter antibiotic lethality [36]. For example, a genetic deletion of enzymes participating in purine biosynthesis exerted significant decreases in ampicillin (AMP) and ciprofloxacin (CIP) lethality in *E. coli* compared to the wildtype. The underlying mechanism is that a decreased abundance of purine triggers a reduction in ATP demand and decreases the toxic metabolic byproducts generated by central carbon metabolism, and the cellular respiration ultimately exacerbates antibiotic-mediated killing. In the present study, all nucleotides and derivatives were significantly downregulated, and a pathway analysis showed that purine metabolism was also significantly enriched. This suggested that nucleotides may also be involved in *H. contortus* IVM resistance. In addition, amino acids are primarily required for nucleotide biosynthesis. This further illustrates that amino acid metabolism may be closely related to IVM resistance in *H. contortus*.

Based on the above discussion, our proposal for the IVM-resistant mechanism in *H. contortus* is that amino acid metabolism may provide resistant strains with specific adaptive traits to counteract the IVM treatment (Figure 6). The decreased amino acids and derivatives suppress the TCA cycle by substrate activation and lead to a reduced concentration of NADH. The NADH is oxidized through the electron transport chain to generate PMF, leading to a decrease in PMF, which is responsible for decreased IVM uptake, ultimately reducing the intracellular IVM concentration. The metabolic state of *H. contortus* thereby influences its susceptibility to IVM. In addition, the active transport of IVM by ABC transporters, which are responsible for IVM elimination, further facilitates the emergence of IVM resistance.

## 5. Conclusions

In the present study, metabolomics was used to analyze the metabolic alterations in an IVM-resistant *H*. *contortus* strain. The results revealed that some important metabolites, such as amino acids and derivatives, may be related to IVM resistance. In conclusion, we propose that *H*. *contortus*’s susceptibility to IVM may be strongly associated with metabolic states and that specific metabolites may alter the susceptibility of *H*. *contortus* to IVM.

## Figures and Tables

**Figure 1 animals-13-00456-f001:**
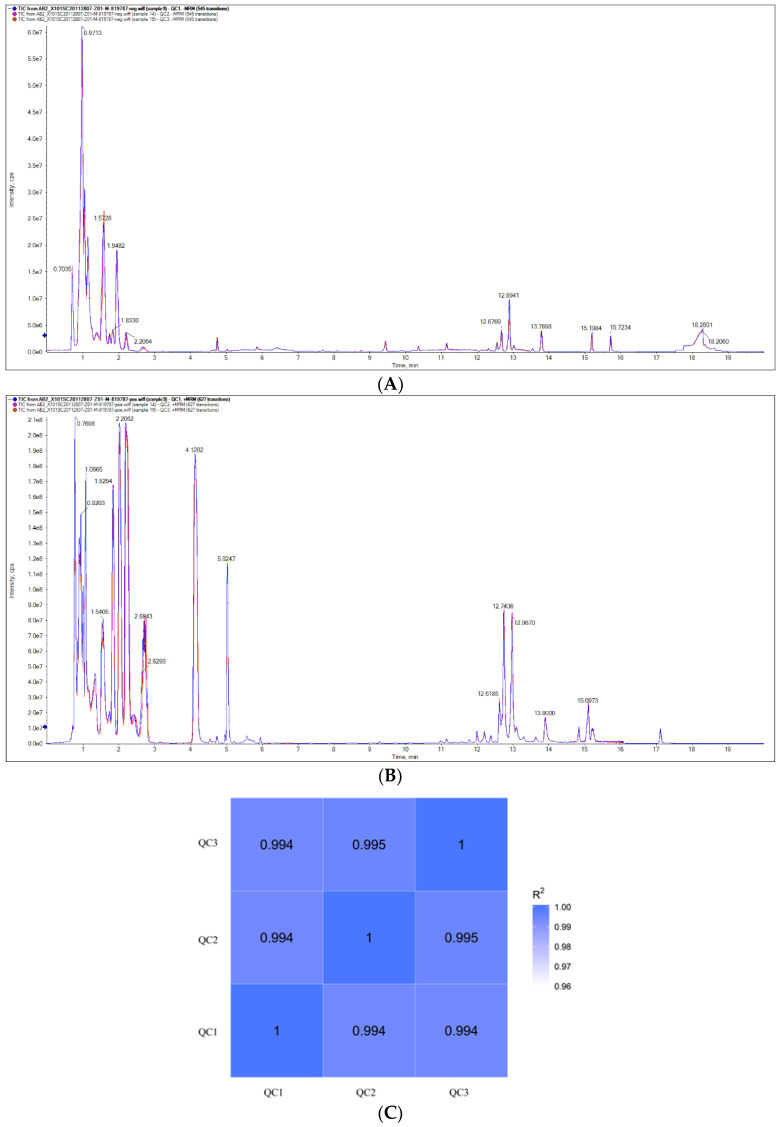
QC sample total ion flow diagrams and Pearson correlations. (**A**) Negative ion mode. (**B**) Positive ion mode. (**C**) Pearson correlations between QC samples.

**Figure 2 animals-13-00456-f002:**
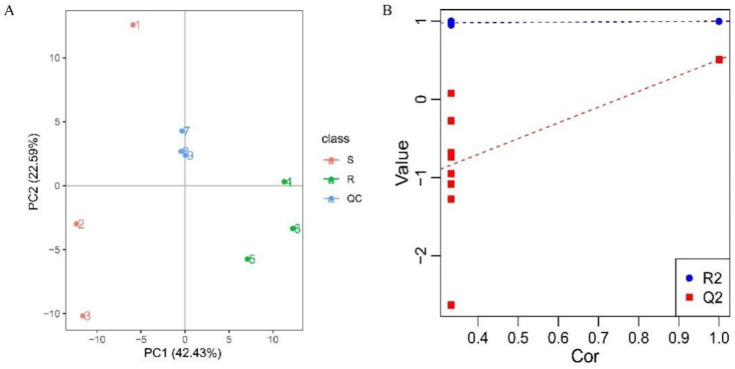
The graphs of the PCA and PLS-DA analyses for the total samples. (**A**) PCA analysis of the IVM-susceptible and -resistant strains. Red circles, susceptible strain; green circles, resistant strain; blue circles, QC samples. (**B**) PLS-DA analysis for the total samples.

**Figure 3 animals-13-00456-f003:**
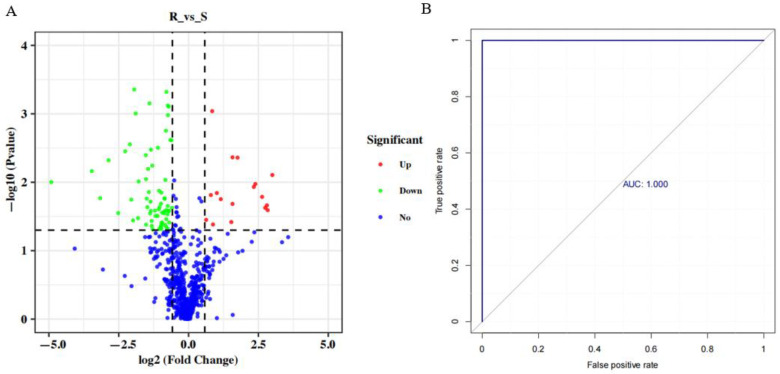
Differential metabolite volcano map and ROC curve of differential metabolites in the IVM-resistant strain compared to the susceptible strain. (**A**) Differential metabolite volcano map of the IVM-susceptible and -resistant strains. Red circles-upregulated metabolites, green circles- downregulated metabolites, dark blue circles-no change metabolites. (**B**) ROC curve of differential metabolites in the IVM-resistant strain compared to the susceptible strain.

**Figure 4 animals-13-00456-f004:**
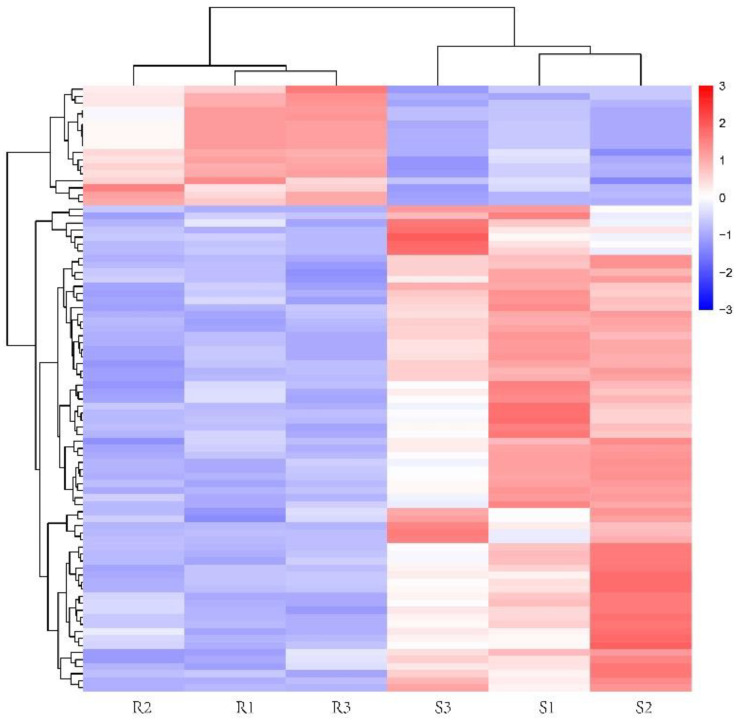
Cluster heat map showing the relative abundance of differential metabolites between the ivermectin (IVM)-resistant and -susceptible strains of *Haemonchus contortus*. R1–3 represent samples from IVM-resistant strains; S1–3 represent samples from IVM-susceptible strains.

**Figure 5 animals-13-00456-f005:**
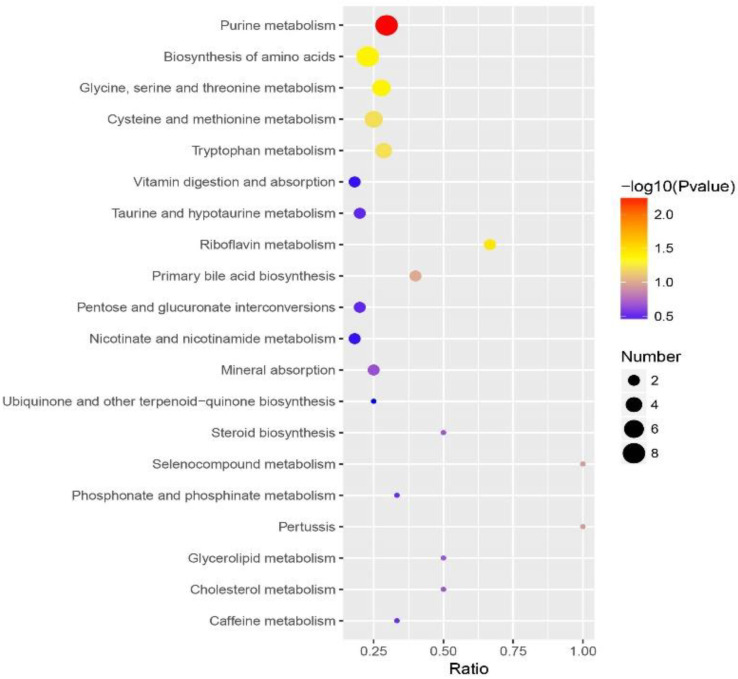
KEGG enrichment analysis of differential metabolites between ivermectin-resistant and -susceptible strains of *Haemonchus contortus*. The *x* and *y* axes represents the enrichment ratio and pathways, respectively. The sizes and colors of the circles represent the number of metabolites and the *p* value, respectively.

**Figure 6 animals-13-00456-f006:**
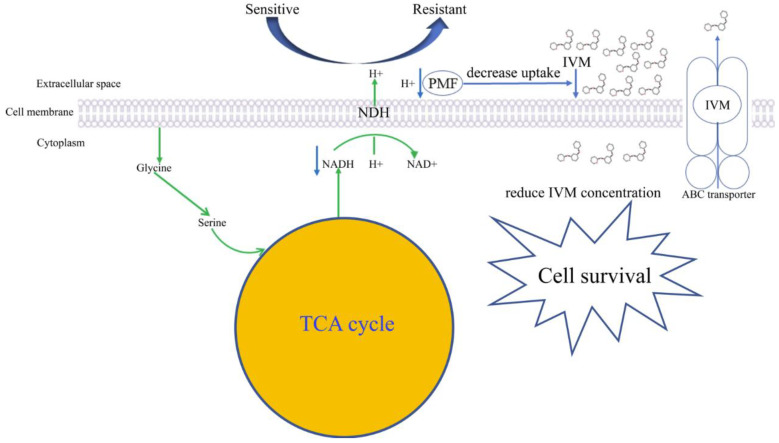
Proposed mechanism for ivermectin resistance in *Haemonchus contortus*.

**Table 1 animals-13-00456-t001:** Quantities and types of total metabolites identified in the ivermectin-resistant and -susceptible strains of *Haemonchus contortus*.

Type	Quantity
Amino Acids and Derivatives	154
Organic Acids and Derivatives	105
Nucleotides and Derivates	74
Fatty Acyls	50
Carbohydrates and Derivatives	42
Hormones	36
Phospholipids	26
Carnitines	22
Organic heterocyclic compounds	20
Sugar Acids and Derivatives	14
Vitamins	13
Eicosanoids	12
Others	11
Bile Acids	11
Benzoic Acid and Derivatives	11
Indole and Derivatives	10
Benzene and substituted derivatives	10
Cholines	9
Pyridine and Derivatives	9
Sugar Alcohols	7
TCA Cycle	7
Phenols and Derivatives	7
Polyamines	7
Phenolamides	5
Prenol lipids	4
Cinnamic acids and derivatives	3
Pyrimidines and pyrimidine derivatives	3
Organic nitrogen compounds	2
Ketones	2
Purines and purine derivatives	2
Glycerolipids	2
Co-Enzyme Factors	2
Alcohols and polyols	2
Benzene and substituted derivatives	2
Amines	2
Esters	1
Organic oxygen compounds	1
Aldehydes	1
Lignans, neolignans, and related compounds	1
Ethers	1
Steroids and steroid derivatives	1
Phenylpropanoids and polyketides	1
total	705

## Data Availability

All data generated during the study are included in the published article(s) cited within the text and Appendix A and acknowledged in the reference section.

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
