# Peer review of "Comparative Metabolome Analyses of Ivermectin-Resistant and -Susceptible Strains of Haemonchus contortus"

_animals, 2023, doi:10.3390/ani13030456_

Round 1

Reviewer 1 Report

The manuscript “Comparative metabolome analyses of ivermectin resistant and susceptible strains of Haemonchus contortus” describes a very interesting study comparing the metabolome of ivermectin-resistant and susceptible H. contortus isolates, showing amino acid metabolism cascades potentially associated with anthelmintic resistance. However, to be suitable for publication, some revision is needed, and more details of Methods and sample collection are required.

Please revise English grammar in the Introduction and Material and Method sections. For example, the following sentences are confusing, it seems that something is missing: “H. contortus infection is one of the major health problems in small ruminants, cause edema, anemia and die suddenly in infected sheep and goats, especially in heavily infected young animals” and “Worms (100 mg) for each strain was prepared and extraction procedures followed a previous study”.

In “The same number of male and female adults (160/sample) of haecon-5-S and zhaosu-R, respectively, were divided into two groups with three biological repeats in each strain”, it seems that it was used males from haecon and females from zhaosu. Make clear the number of males and females in each strain, the number of samples and the number of replicates.

I know that promote your study is essential, but please avoid overselling it, as in “From the time of first used, nematodes control in all small ruminants relies widely on the use of IVM.” and “… understanding IVM resistance mechanisms remains the most urgent unmet clinical need in anthelmintic treatment”.

The drug resistance reported for cancer cells may have given insights for investigating the same pathways in H. contortus. However, as it was not the subject of the study, it should not be mentioned in the Abstract and Simple Summary.

Provide detailed methods for host infection (the use of goats was mentioned only on Author’s contribution), adult H. contortus collection and how the samples were handled and preserved after collection. Which information or data were used to confirm that the strains are resistant or susceptible to IVM? Mention if adults were collected after host exposition to IVM or not, because based on what is stated in Conclusion (“… metabolomics was used to analyze the metabolic alteration in H. contoruts IVM-resistant strain originating from IVM-induced selective pressures”), it seems that IVM treatment was used.

Define VIP and FC in Methods. Why was it used FC≥1.5 or FC≤0.667?

The quality control samples should be mentioned in the Methods section. Move the paragraph (lines 112-115) from Results to Methods and provide details about QC samples 1, 2, and 3. The results of QC and instrument calibration, including Figures, should be presented in Supplementary material.

The paragraph about PCA (lines 132-133) should be moved to Methods section. In methods, include details and parameters used for PCA and PLS-DA analyses.

There is much information in Figure 4, I suggest using the data from the Figure (and not from the Supplementary material) in a table, separating the number of metabolites of each category by susceptible and resistant H. contortus strains.

No result of sequencing was presented in the manuscript, contrary to what was stated in “These results further confirm the high accuracy and reproducibility of the sequencing data”.

Despite being considered an antibiotic, IVM is mainly used as an anti-parasitic drug, and as the study tests it in H. contortus, the use of the term antibiotic (in Introduction and Discussion section) must be contextualized.

Author Response

We are grateful to the reviewer for their effort, time and constructive reports. In the word file are our responses to the issues raised by the reviewer.thank you

Author Response

We are grateful to the reviewer for their effort, time and constructive reports. In the word file are our responses to the issues raised by the reviewer.

Round 2

Reviewer 2 Report

The authors sufficiently addressed the major problems I had, and fixed the minor issues.  I believe the manuscript is ready for publication.